# Characterization of Serum Cytokine Profiles of Patients with Active Lupus Nephritis

**DOI:** 10.3390/ijms241914883

**Published:** 2023-10-04

**Authors:** Zahrà Rahmé, Chiara Franco, Claudio Cruciani, Federico Pettorossi, Alice Zaramella, Stefano Realdon, Luca Iaccarino, Giulia Frontini, Gabriella Moroni, Andrea Doria, Anna Ghirardello, Mariele Gatto

**Affiliations:** 1Unit of Rheumatology, Department of Medicine, University of Padova, 35128 Padova, Italy; zahra.rahme@studenti.unipd.it (Z.R.); chiara.franco@aopd.veneto.it (C.F.); claudio.cruciani94@gmail.com (C.C.); luca.iaccarino@unipd.it (L.I.); adoria@unipd.it (A.D.); anna.ghirardello@unipd.it (A.G.); 2Department of Surgery, Oncology and Gastroenterology (DISCOG), University of Padova, 35128 Padova, Italy; 3Veneto Institute of Oncology IOV-IRCCS, 35128 Padova, Italy; 4Oncology Referral Center of Aviano (CRO)-IRCCS, 33081 Aviano, Italy; stefano.realdon@cro.it; 5Nephrology and Dialysis Unit, San Paolo Hospital, 20153 Milan, Italy; freegiuly92@gmail.com; 6Nephrology and Dialysis Division, IRCCS Humanitas Research Hospital, Rozzano, 20089 Milan, Italy; 7Unit of Rheumatology, Department of Clinical and Biological Sciences, University of Turin, 10124 Torino, Italy

**Keywords:** lupus nephritis, cytokines, IL-37, IL-10, BAFF, renal response, biomarkers

## Abstract

Cytokines contribute to the pathogenesis of lupus nephritis (LN), yet their value as prognostic biomarkers is still debated. We aimed to describe the serum cytokines’ profiles and prospectively assess correlations with disease features and renal response in a multicentric cohort of consecutive adult patients with biopsy-proven active LN. Cytokine associations with clinical and serological data were performed at LN diagnosis (T0), and at 3 (T3) and 6 months (T6) of follow up. Renal response according to EULAR definition was assessed at T3, T6 and T12. BAFF and interleukin (IL)-37 were measured by ELISA; IL-2, IL-10, IL-17A and IL-18 by a bead-based multiplex cytokine assay (Luminex). Thirty-nine patients with active LN (age 40.5 ± 15.6 years; F 71.8%; 84.6% proliferative LN) were enrolled, of whom twenty-nine displayed complete longitudinal records. At T0, we observed higher levels of IL-37 and IL-17 in proliferative vs. non-proliferative LN (IL-37: 0.0510 (0.0110–0.2300) vs. 0.0000 (0.0000–0.0397) ng/mL, *p* = 0.0441; IL-17: 2.0920 (0.5125–17.9400) vs. 0.0000 (0.0000–0.6025) pg/mL, *p* = 0.0026, respectively), and positive correlations between IL-10 and 24 h proteinuria (r = 0.416, *p* = 0.0249) and anti-dsDNA levels (r = 0.639, *p* = 0.0003). BAFF was higher in patients with low complement (*p* < 0.0001). We observed a sustained correlation between BAFF and IL-10 throughout T6 (r = 0.654, *p* = 0.0210). Higher baseline IL-37 and BAFF levels were associated with renal response at T3 and T6, respectively, while baseline IL-18 levels were higher in patients achieving response at T12. Our study highlights the complexity of the cytokine network and its potential value as a marker of active LN and renal response.

## 1. Introduction

Systemic lupus erythematosus (SLE) is a systemic autoimmune disease characterized by the production of autoantibodies targeting nuclear and cytoplasmic antigens. Organ involvement covers a broad spectrum of manifestations, frequently affecting skin, blood, joints and even noble organs such as kidneys and the brain, with potential permanent organ damage [1].

The pathogenesis of SLE is complex and not yet defined. As in other autoimmune diseases, the underlying mechanisms involve a loss of immune tolerance in predisposed individuals as a result of environmental stochastic events and infectious triggers. The dysregulated mechanisms are then sustained by the interaction between innate and adaptive immunity [2].

Lupus nephritis (LN) is one of the most frequent and severe manifestations of SLE, affecting around 40–70% of SLE patients, leading to organ damage up to end-stage kidney disease (ESKD) [3].

Among LN pathogenic mechanisms, abnormalities in apoptosis and clearance of debris play a role, resulting in the release of nuclear and cytoplasmic material that activates innate immunity, leading to the secretion of pro-inflammatory molecules activating self-reactive B and T cells, which perpetrate damage through complex pathways [4,5]. Circulating immunomodulatory cytokines are capable of mediating tissue damage, either directly or through the stimulation of autoreactive B and T cells. Recent data have suggested a recurrent panel of cytokines, encompassing B-Lymphocyte stimulator (BlyS or BAFF), interleukin (IL)-37, IL-2, IL-10, IL-17A and IL-18, that are likely involved in LN onset and progression [4,6,7,8,9,10,11]. 

Molecules such as IL-37 and IL-10 mainly exert an anti-inflammatory effect [12,13], while IL-2, IL-17A, IL-18 and BAFF are known to be mostly pro-inflammatory [8,14,15,16,17] and represent actual or potential therapeutic targets in clinical practice.

IL-37 is produced by immune and non-immune cells in response to pro-inflammatory stimuli and observed to be higher in patients with SLE, especially those with renal and mucocutaneous disease [12], while IL-10 seems to play a dual-sided cytokine role in the pathogenesis of SLE [13].

IL-2 affects T cell regulation, shifting the balance between pro-inflammatory effector T cells and regulatory T cells, overall being skewed towards a pro-inflammatory effect in SLE [8]. BAFF is widely implied in B cell survival and maturation, also being involved specifically in the sustainment of local renal inflammation [17]. IL17a stimulates resident cells to produce pro-inflammatory molecules such as IL-6 and IL-8 and attracts monocytes, neutrophils and T cells, which foster local damage [14], and IL-18 stimulates interferon-γ production [15,16].

The aim of our study was to assess the role of key biomarkers of the LN course in a multicenter cohort of SLE patients with active renal involvement.

## 2. Results

### 2.1. Study Population

Thirty-nine patients with biopsy-proven active LN (females *n* = 32 (71.8%); age 40.5 ± 15.6 years) were recruited at the time of active LN diagnosis. Among them, 33 (84.6%) displayed proliferative LN, encompassing 2 (5.1%) Class III LN, 26 (66.7%) Class IV LN and 5 (7.7%) mixed-class LN.

Twenty-nine (74.4%) out of thirty-nine patients (females *n* = 19 (57.6%), age 39.7 ± 16.7 years) were prospectively followed up for 52.3 (15.5–100.8) months and had their serum cytokines and clinical characteristics assessed at baseline (T0) and longitudinally. 

Demographic and clinical features of the prospective cohort are reported in Table 1.

### 2.2. Cross-Sectional Study 

Cytokine values at baseline (T0) are shown in Appendix A.

Statistically significant correlations were found between BAFF and IL-10 (r = 0.421, *p* = 0.0183), IL-10 and IL-18 (r = 0.351, *p* = 0.0265), IL-2 and IL-17A (r = 0.384, *p* = 0.014) and BAFF and IL-18 (r = 0.342, *p* = 0.055) (Figure 1).

Considering patients displaying measurable levels of the whole cytokine panel, IL-37 and IL-17A were significantly higher in patients with proliferative LN vs. non-proliferative LN (IL-37: 0.0510 (0.0110–0.2300) vs. 0.0000 (0.0000–0.0397) ng/mL, *p* = 0.0441; IL-17A: 2.0920 (0.5125–17.9400) vs. 0.0000 (0.0000–0.6025) pg/mL, *p* = 0.0026) (Figure 2A,B). 

Patients with proliferative classes were stratified according to proteinuria levels (>3 g/day (*n* = 12) vs. ≤3 g/day (*n* = 17)). Levels of cytokines did not differ significantly between the two groups (IL-37, *p* = 0.573; BAFF, *p* = 0.865; IL-2, *p* = 0.727; IL-10, *p* = 0.782; IL-17, *p* = 0.235; IL-18, *p* = 0.914).

The correlation between serum cytokines and LN-related clinical and laboratory items was statistically significant between IL-10 and 24 h proteinuria (r = 0.416, *p* = 0.0249) (Figure 3A), between IL-10 and anti-dsDNA levels (r = 0.639, *p* = 0.0003) (Figure 3B) and between IL-17A and C3 levels (r = −0.378, *p* = 0.0474) (Figure 3C).

BAFF levels were significantly increased in patients with low complement levels (1.5980 (0.8350–2.7400) vs. 0.0750 (0.0000–0.3260), *p* < 0.0001).

Although borderline for statistical significance, IL-10 showed a trend for inverse correlation with C3 and C4 levels (r = −0.359, *p* = 0.0608 and r = −0.326, *p* = 0.0906, respectively). 

No significant correlation was found between SLEDAI-2K score and serum cytokines.

### 2.3. Prospective Study

Values of serum cytokines levels at 3-month (T3) and 6-month (T6) timepoints are shown in Appendix A. IL-37 and BAFF showed a significant decrease from baseline throughout T6 (Appendix A and Figure 4).

At T3, significant correlations were found between IL-2 and IL-17A (r = 0.627, *p* = 0.001), BAFF and IL-10 (r = 0.578, *p* = 0.0061) and between IL-10 and IL-18 (r = 0.429, *p* = 0.0363).

At T6, a statistically significant correlation was found between BAFF and IL-10 (r = 0.654, *p* = 0.0210) and between IL-18 levels and SLEDAI-2K score (r = −0.643, *p* = 0.0241).

No significant correlations nor associations were found between serum cytokines and other clinical or laboratory features at T3 and T6.

### 2.4. Renal Response

Associations between baseline serum cytokine levels and renal response at 3, 6 and 12 months were assessed. Out of the 29 patients followed up over time, 19 (65.6%) reached renal response at T3 (PRR = 15, CRR = 4), and 20 (68.9%) at T6 (PRR = 12, CRR = 8) who maintained it throughout T12 (PRR = 10, CRR = 10).

Baseline levels of IL-37 and IL-17 were significantly higher in responders vs. non-responders at T3 (IL-37: 0.0990 (0.0282–0.2325) vs. 0.0000 (0.0000–0.0125), *p* = 0.0058; IL-17: 1.3200 (0.7300–21.8100) vs. 0.1800 (0.0000–3.4430, *p* = 0.038) (Figure 5A,B). 

Baseline levels of IL-37 and BAFF were higher in responders vs. non-responders at T6, though IL-37 was borderline for statistical significance (IL-37: 0.0870 (0.0180–0.2300 vs. 0.0050 (0.0000–0.0290), *p* = 0.065 and BAFF: 1.3560 (1.1070–3.9100) vs. 0.8130 (0.5930–1.7170), *p* = 0.012) (Figure 5C,D). 

Baseline levels of IL-18 and BAFF were higher in responders vs. non-responders at T12, though BAFF was borderline for statistical significance (IL-18: 313.9 (205.2–413.9) vs. 213.4 (16.29–281.8), *p* = 0.047 and BAFF: 1.2850 (0.7630–2.6010) vs. 1.3230 (0.7295–2.8140), *p* = 0.062) (Figure 5E,F).

No independent predictors of renal response were found among baseline cytokine levels.

## 3. Discussion

We investigated the profile of cytokines involved in the pathogenesis of LN with immunomodulatory roles, encompassing BAFF, IL-2, IL-37, IL-10, IL-17A and IL-18.

At the time of active LN, correlations were found between serum cytokine levels and items commonly used in clinical practice to monitor disease progression. Levels of IL-10 at baseline correlated with 24 h proteinuria, anti-dsDNA antibody levels and CRP, while IL-17A and BAFF were found to be higher in patients with low complement levels. Additionally, we observed correlations between BAFF and IL-10, BAFF and IL-18, and IL2 and IL-17A. The correlation between BAFF and IL-10 was confirmed over time throughout T6, reflecting the close link between these cytokines. In fact, several studies have demonstrated the key role of both BAFF and IL-10 in sustaining inflammation in LN. BAFF has been shown to be produced by glomerular macrophages and mesangial cells and to promote B cell activation both directly and through the induction of pro-inflammatory molecules such as interferon α (IFN-α), thereby sustaining a positive autoreactive loop within the kidney microenvironment [4,6,18,19,20]. IL-10 has been hypothesized to play a dual role in SLE and LN [21]. While IL-10 is known to display regulatory capabilities, its levels are increased in patients with SLE, even during pregnancy, being associated with higher disease activity [21,22]. Conversely, the abatement of BAFF levels through the use of belimumab was suggested to incite new-onset LN in selected SLE patients, likely through the concomitant decrease in IL-10 [9,23,24,25,26]. In our cohort, the two cytokines appear tightly linked, suggesting a possible counterbalance in vivo which may modulate the immune reactivity over time.

Comparing serum cytokine levels in subjects with proliferative vs. non-proliferative LN, significantly increased levels of both IL-37 and IL-17A were observed in patients with proliferative classes. IL-37 is a regulatory cytokine with suppressive capabilities which is produced by peripheral blood mononuclear cells (PBMCs), dendritic cells (DCs), epithelial cells, macrophages and T cells in response to pro-inflammatory stimuli [27]. Interestingly, IL-37 was shown to be higher in patients with LN and mucocutaneous activity [7,28], and previous studies documented a correlation between IL-37 mRNA overexpression and risk of developing LN in patients with SLE [29,30], thus posing the issue of IL-37 being involved in SLE progression. Importantly, sustained increased levels of IL-37 seem to lose their regulatory capability due to the formation of inactive homodimers or to sequestration of its soluble receptor which is shared between IL-37 and IL-18 [31], though a rebound secretion cannot be excluded, similarly to what is hypothesized for IL-10. 

IL-17 is a pro-inflammatory cytokine secreted mainly by Th17 cells, particularly in its IL-17A form [32]. IL-17 can sustain LN both through recruitment of T cells to the kidneys, and by activating B cells to secrete autoantibodies. Previous studies on patients with LN showed abnormal levels of IL-17A both in serum and urine; importantly, serum IL-17A levels heralded major kidney damage in patients with class IV or V glomerulonephritis [10,26,33,34,35,36], in keeping with our data.

While no independent predictors of renal response were found, we observed a significant association between baseline IL-37 levels and renal response at three months, with a trend towards significance in patients attaining response at six months. Increased baseline BAFF levels were associated with renal response at six and twelve months. Interestingly, patients with delayed renal response showed increased IL-18 at baseline. IL-18 is a pro-inflammatory cytokine released mainly by epithelial and innate immunity cells, which stimulates cell survival and lymphocytic maturation, particularly of Th1 cells, and the production of pro-inflammatory cytokines [16]. Through interaction with its receptor, it is able to stimulate IFN-gamma production and local disease damage. Previous studies showed that higher serum levels of IL-18 were observed during disease flares and in subjects with active LN compared to subjects affected with SLE without renal involvement, and correlations between IL-18 and anti-dsDNA levels and proteinuria were shown [11,37,38,39,40,41]. Increased baseline IL-18 levels, similar to increased BAFF levels, could signify an active disease which is prone to respond to immunosuppressive treatment.

Thus, 24 h proteinuria was shown to correlate with cytokines related with higher disease activity such as IL-10 and IL-18, likely mirroring the renal involvement in patients with a more pro-inflammatory cytokine pattern. 

Although we stratified patients with proliferative forms of nephritis according to proteinuria levels (>3 g/day vs. ≤3 g/day), levels of cytokines did not differ between the two; given the small numbers in this cohort, no firm conclusions can be drawn on the role of proteinuria itself as a possible predictor of cytokine imbalance, requiring a larger cohort.

Major limitations of this study encompass the limited sample size, which does not allow solid inference on cytokine effects on LN course, the lack of cytokine measurements on renal tissue and in healthy controls. To a lesser extent, other limitations concern the lack of unconventional biomarkers in routine analysis such as anti-C1q antibodies and anti-nucleosome antibodies to better profile LN patients.

Altogether, our data highlight the complexity of the cytokine network in LN and the dual-faced role of IL-37 and IL-10 during systemic inflammation, which should be carefully considered in order to exploit the cytokine network as a potential tool for patient monitoring. 

## 4. Materials and Methods

### 4.1. Study Population

Consecutive adult (≥18 years old) patients with biopsy-proven LN followed up at the Unit of Rheumatology of Padova University Hospital and at IRCSS Humanitas Research Hospital, Milano, Italy were enrolled in the study.

Criteria for inclusion were: (i) SLE classified according to EULAR [42] or ACR criteria [43]; (ii) active biopsy-proven LN; (iii) at least two evaluations per year; and (iv) substantial data availability at subsequent evaluations. Subjects with major comorbidities (i.e., infections, malignancies), pregnancy, or patients requiring renal replacement therapy were not considered for the study.

LN was classified according to the recent revision of International Society of Nephrology/Renal Pathology Society (ISN/RPS) criteria [44]; class III, IV and mixed forms with a class IV component were considered proliferative.

### 4.2. Study Design

This is an observational study entailing both a cross-sectional (at baseline) and a prospective phase which was conducted on a multicentric cohort of SLE patients with active, biopsy-proven LN both considering new onsets and flares.

#### 4.2.1. Cross-Sectional Phase

Cytokine serum levels were measured at the time of diagnosis of active LN (baseline: T0) and associations with baseline clinical and laboratory features were assessed. Among clinical features, we considered renal function, hypertension and renal items at the SLEDAI-2K score, as well as renal histology. Among laboratory features, we considered complement levels, anti-dsDNA antibody levels and hematological features.

#### 4.2.2. Prospective Phase

Associations of cytokine serum levels with clinical and laboratory features were assessed at 3 (T3) and 6 (T6) months after the diagnosis of active LN.

Samples were collected at 3-month (T3) and at 6-month (T6) timepoints. 

Renal outcome was assessed at early timepoints (T3 and T6) and at 12 months (T12) after the diagnosis of active LN. Association and potential predictive value of baseline cytokine levels and renal outcome were investigated at each timepoint.

Renal response was defined according to European recommendations for LN [45]. Patients were considered responders if they achieved either a partial (PRR) or complete renal response (CRR). CRR was defined as proteinuria < 0.5–0.7 g/24 h (with GFR normalization/stabilization) by 12 months. PRR was considered as evidence of improvement in proteinuria (with GFR normalization/stabilization) by 3 months, and at least 50% reduction in proteinuria (partial clinical response) by 6 months. Patients not meeting any of these criteria were considered non-responders.

### 4.3. Cytokines and Serology Analysis

The levels of serum human BAFF and IL-37 (ng/mL) were measured through a commercially available solid-phase enzyme-linked immunosorbent assay (ELISA) using Quantikine BAFF ELISA kit (R&D, Systems, Inc., Minneapolis, MN, USA, detection limit (DL) 0.02 ng/mL), and IL-37 ELISA kit (Adipogen Life Sciences, Liestal, Switzerland, DL 0.001 ng/mL) following the manufacturer’s recommendations. Cytokine levels were measured in duplicate. Briefly, diluted serum (1:2) and standards were placed into plate wells pre-coated with anti-human BAFF monoclonal antibody or anti-human IL-37 polyclonal antibody. The plates were incubated for 3 h at room temperature (BAFF) or overnight at 4 °C (IL-37) on a microplate shaker, and then washed. Followed by addition of Detection Antibody (Adipogen) on the IL-37 plate, it was incubated for 1 hour at 37 °C. Horseradish peroxidase-conjugated antibodies were added to each well and the plate was incubated for 1 h at room temperature on the shaker, and then washed. Substrate solution was added to each well and the plates were incubated for 30 min (BAFF) or 10 min (IL-37) at room temperature to allow color development. Stop solution was then added and the absorbance was read at 450 nm by a Multiskan EX microplate reader (Labsystems Italia, Milano, Italy). The standard curve was interpolated in a quadratic equation. 

Human IL-17A, IL-18, IL-10 and IL-2 (pg/mL) were measured by a multiplexed fluorescent bead-based immunoassay using Luminex® technology Milliplex® Human Cyto Panel A 4-plex HCYTA-60K-05® (Merck Millipore, Billerica, MA, USA, DL 0.6 pg/mL), according to the manufacturers’ instructions. Briefly, serum samples were centrifuged at 13,000 rpm for 10 min to exclude debris. Wash buffer was added into each well of the plate, and the plate was mixed on a plate shaker for 10 min at room temperature and then washed. Undiluted serum, standards and controls were placed into the appropriate wells. Magnetic beads were added into the wells and the plate was incubated with agitation on a plate shaker overnight at 2–8 °C. After the plate washing, the detection antibodies were added into the wells, and the plate was incubated on a plate shaker for 1 h at room temperature. Streptavidin–Phycoerythrin was added to each well, and the plate was incubated on a plate shaker for 30 min at room temperature and then washed. Sheath Fluid PLUS was added to all wells and the magnetic beads were resuspended on a plate shaker for 5 min. The Median Fluorescent Intensity (MFI) data was analyzed using a 5-parameter logistic curve fitting method to calculate analyte (IL-17A, IL-18, IL-10 and IL-2) concentrations simultaneously in each serum sample using Luminex xMAP® technology (Luminex Corporation, Austin, TX, USA) and Luminex xPONENT 3.1 Software (Luminex Corporation, Austin, TX, USA).

Serology laboratory markers, such as anti-dsDNA, were measured at the University-Hospital Central Laboratory using a Chemiluminescence ImmunoAssay (CLIA) assay.

### 4.4. Ethics

This study was conducted in accordance with the Declaration of Helsinki and approved by the local ethics committee (protocol code 5349/AO/22). All participants gave written informed consent to study enrolment and data analysis. 

### 4.5. Statistics

Demographic features and serology were expressed as mean ± standard deviation (SD) and results of cytokine levels as median (IQR) for continuous variables, while categorical data were expressed as numbers and percentages. Comparisons between groups was carried out by the Mann–Whitney U test or analysis of variance with Friedman’s correction for more than two groups, and Spearman’s correlation coefficient for correlations between cytokines and between cytokines and clinical items. The distribution of data concerning cytokines was to be considered non-normal. Logistic regression analysis was performed to assess independent predictors of response. A two-tailed *p*-value (*p*) ≤ 0.05 was considered statistically significant. Data were analyzed using GraphPad Prism® version 9 and Statistics Pacytokinesage for Social Sciences (SPSS) for Windows, version 22.0, SPSS Inc., Chicago, IL, USA.

## 5. Conclusions

Our findings support the role of BAFF, IL-37, IL-2, IL-10 and IL-18 in the pathogenesis of LN.

Cross-sectional data show that proliferative LN classes display a distinct cytokine pattern and inform the close relationship between BAFF, IL-10 and IL-18 in active LN. Longitudinal observations underline the potential role of cytokines as a complementary tool to anticipate disease course. The most relevant findings show increased baseline levels of IL-37 and BAFF in patients who achieved renal response after 3 and 6 months of treatment, while elevated IL-18 differentiated delayed responders from non-responders.

The reasons behind cytokine variations and their inter-relationships require further studies which may help to shed light on prominent mechanisms in different stages of disease. 

In summary, our study highlights the value of serum cytokines as biomarkers of active LN and renal response.

## Figures and Tables

**Figure 1 ijms-24-14883-f001:**
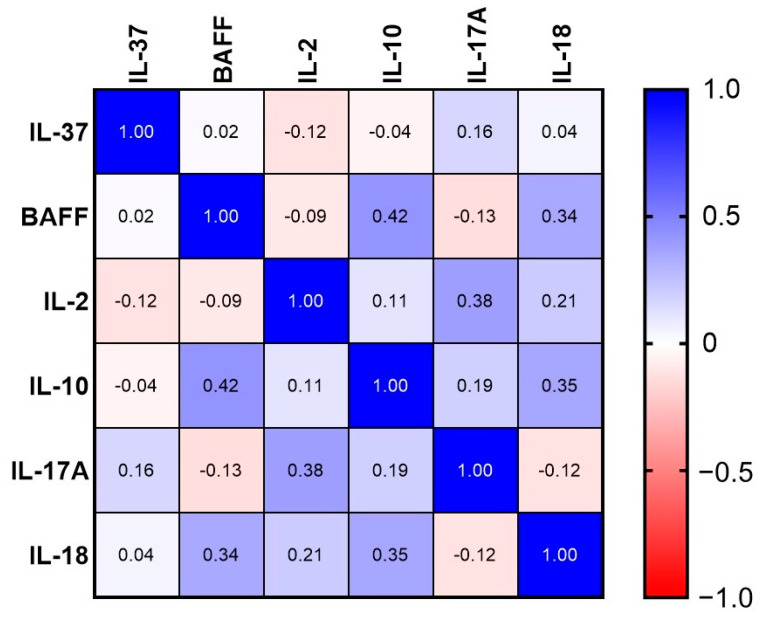
Heatmap reporting Spearman R values referred to correlations between serum cytokine levels at baseline (T0).

**Figure 2 ijms-24-14883-f002:**
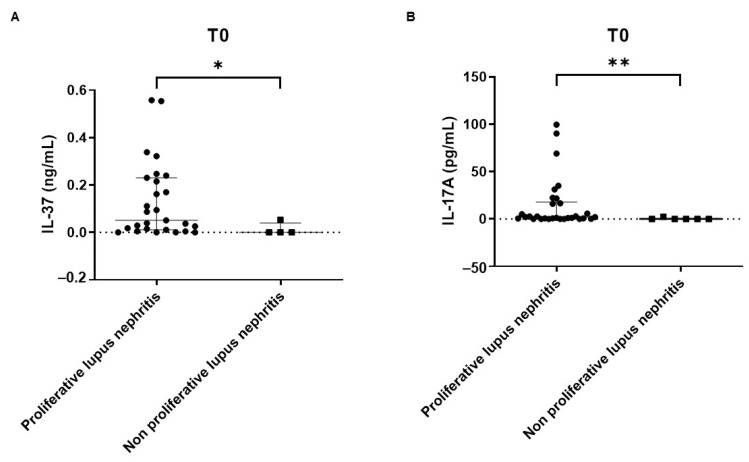
Serum levels of IL-37 (**A**) and IL-17A (**B**) in patients with proliferative LN vs. those with non-proliferative LN at T0. * *p* < 0.05; ** *p* < 0.01; black circle: proliferative lupus nephritis patients; black square: non proliferative lupus nephritis patients.

**Figure 3 ijms-24-14883-f003:**
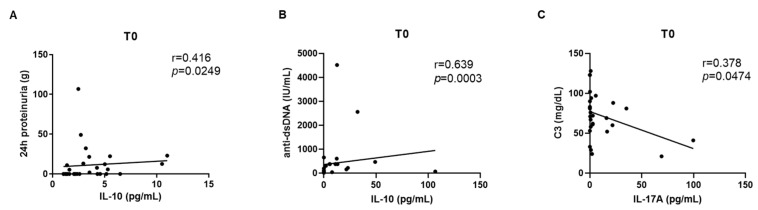
Correlation at T0 between IL-10 vs. 24 h proteinuria (**A**); IL-10 vs. anti-dsDNA (**B**); IL-17A and C3 at T0 (**C**).

**Figure 4 ijms-24-14883-f004:**
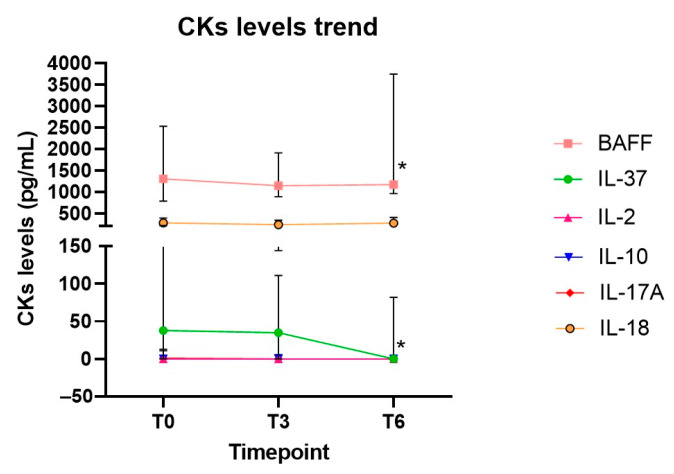
Variations of cytokines levels (median and error, IQR) at T0, T3, T6. * *p* < 0.05.

**Figure 5 ijms-24-14883-f005:**
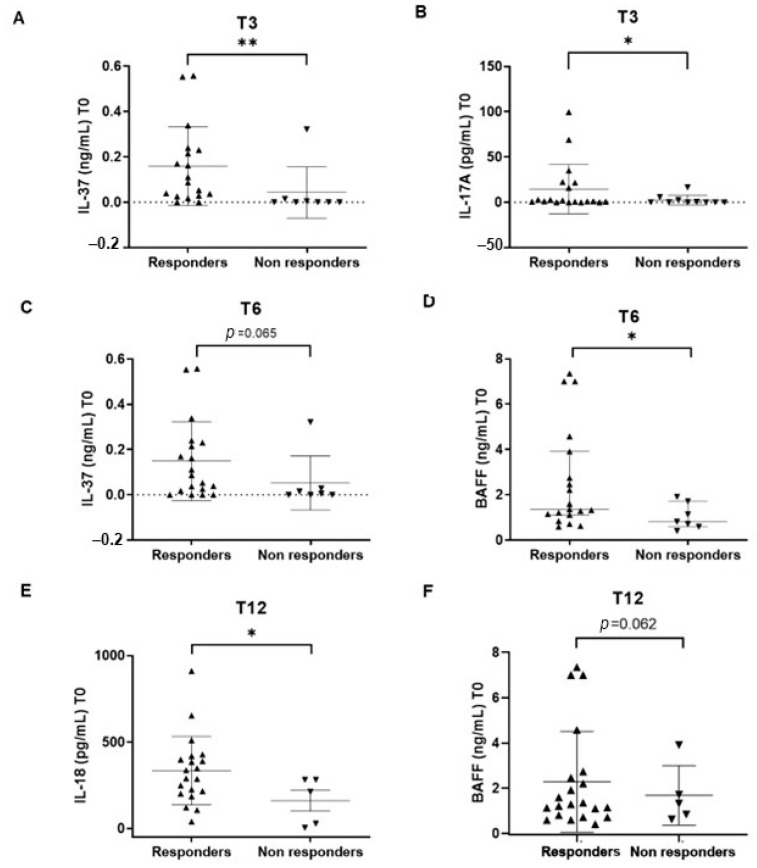
IL-37 (**A**) and IL-17A (**B**) serum levels at baseline in responders vs. non-responders at T3, BAFF (**C**) and IL-37 (**D**) serum levels at baseline in responders vs. non-responders at T6. IL-18 (**E**) and BAFF (**F**) serum levels at baseline responders vs. non-responders at T12. Mann–Whitney U test. * *p* < 0.05; *p* < 0.01; black triangles pointing upwards: responders patients; black triangles pointing downwards: non responders patients.

**Table 1 ijms-24-14883-t001:** Baseline demographic, clinical and serological features of lupus nephritis patients followed-up prospectively.

Demographics	
Patients, *n*	29
Age at baseline, mean ± SD (years)	39.7 ± 16.7
Females, *n* (%)	19 (65.5)
Caucasians *n* (%)	24 (82.8)
Smokers, *n* (%)	5 (17.2)
SLE disease duration at baseline, mean ± SD (years)	9.9 ± 9.2
**Histological Class at Renal Biopsy**	
Proliferative LN, *n* (%)	24 (82.8)
Non-proliferative LN, *n* (%)	5 (17.2)
Class II	2 (6.9)
Class III	2 (6.9)
Class IV	20 (69.0)
Class V	2 (6.9)
Mixed class (III + IV, IV + V, III + V)	2 (6.9)
Other (podocytopathy)	1 (3.4)
**Clinical Features at Baseline**	
Chronic kidney disease, *n* (%)	5 (17.2)
End-stage kidney disease, *n* (%)	0 (0)
Serology, *n* (%)	
ANA > 1:80	29 (100)
Anti-dsDNA	22 (75.9)
Anti-Sm	8 (27.6)
Anti-U1RNP	6 (20.7)
Anti-Scl70	5 (17.2)
Anti-SSA	18 (62.1)
Anti-SSB	9 (31.0)
Anti-phospholipid antibodies	
- Single positivity	9 (31.0)
- Double positivity	2 (6.9)
- Triple positivity	2 (6.9)
Active urinary sediment, *n* (%)	22 (75.9)
Proteinuria > 3 g/24 h, *n* (%)	13 (44.8)
Low complement, *n* (%)	21 (72.4)
Anti-dsDNA antibody levels (UI/mL) (mean ± SD)	420.1 ± 940.4
Hypertension, *n* (%)	14 (48.3)
Anemia, *n* (%)	16 (55.17)
Leucopenia, *n* (%)	11 (25.6)
Thrombocytopenia, *n* (%)	9 (31.0)
SLEDAI-2K score, mean ± SD	10.6 ± 5.0
CRP (mg/L), mean ± SD	2.3 ± 2.7
ESR (mm/h), mean ± SD	24.2 ± 16.3
**Treatment Administered at Baseline**	
CS pulses, *n* (%)	24 (82.8)
Oral CS, *n* (%)	26 (89.7)
MMF, *n* (%)	26 (89.7)
MMF + CNI, *n* (%)	1 (3.4)
IV CYC, *n* (%)	5 (17.2)
RTX, *n* (%)	1 (3.4)
HCQ, *n* (%)	22 (75.9)

CS = corticosteroids; MMF = mycophenolate mofetil; CNI = calcineurin inhibitors; IV CYC = intravenous cyclophosphamide; RTX = rituximab; HCQ = hydroxychloroquine.

## Data Availability

Data are available upon reasonable request from European centers to the corresponding author.

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
