# Peer review of "Characterization of Serum Cytokine Profiles of Patients with Active Lupus Nephritis"

_ijms, 2023, doi:10.3390/ijms241914883_

Round 1

Reviewer 1 Report

The Authors aimed to analyze serum cytokines profile in patients with active lupus nephritis. The study conducted by the Authors is clinically very important and justified as LN is one of the most frequent and severe manifestations of SLE. All the CKs were analyzed in other studies however combination of these biomarkers and prospective analysis might give a clinically useful results.

But I have several concerns to the current study:

  • There were  39 patients included into the study and finally the Authors based only on 29 patients who are additionally divided into groups. The number of patients is really small so the statistics is very weak which makes it impossible to obtain reliable results.
  • There should be provided values of common inflammatory markers like CRP and ESR and related to CKs especially those with proinflammatory properties
  • Only 22 patients were positive for anti-dsDNA antibodies which might be surprising as all of them represent high activity of the disease. Authors should add a comment on it. A good option in clinical practice is assessment of anti-nucleosome antibodies, which also are specific and pathogenic for LN and of great importance especially in patients negative for anti-dsDNA. Another antibodies important in LN are anti-C1q so it is surprising that they were not assessed as the analyzed serological profile was quite wide and in my opinion not fully justified.
  • ANA results need to be supplemented.
  • High frequency of anti-Scl70 is puzzling. Of course they happen in SLE but usually in low titers and the prevalence also is rather low (less than 5%). Methodological aspects can influence the results. Authors should comment on it.
  • Methods for laboratory markers, including serology, should be described.
  • It would be important to get results of CKs in controls to calculate the cut-off levels as well as to compare LN with lupus patients without kidney involvement to establish the role of CKs in LN. Authors may consider this for the future studies.

In my opinion English language is quite good and needs minor improvement.

Author Response

Dear Reviewer,
Thank you very much for reviewing our manuscript.

You can find attached the file containing the detailed responses to the comments and the corresponding revisions and corrections highlighted in the resubmitted manuscript. Please see the attachment.

Reviewer 2 Report

This is an interesting paper. Authors aimed to describe selected serum cytokines profile and prospectively assess correlations with disease features and renal response in patients with biopsy-proven active LN.

To improve the quality of the paper I suggest;

1.      To describe selected population in more detail.

Particularly in the context of comorbidities, that could bias the results (i.e. infections, malignancies). Author state (line 218) that consecutive LN patient were included. Were they “all” consecutive patients? No exclusion criteria were applied? If patients had concurrent infection or malignancy or diabetes or were pregnant this could influence the cytokine profile. In such case additional analysis should be performed, reported and discussed.

2.      To address the question: what was the impact of degree of proteinuria on the cytokines profile?

Authors conclude that proliferative LN classes display a distinct cytokines profile (line 273). If 44% of patients had nephrotic proteinuria, whereas 83% had proliferative LN and proteinuria itself is associated with cytokine production it would be interesting for the readers to know the results of additional analysis of cytokines in the context of proteinuria.

3.      To discuss the study limitations.

Author Response

(The authors gave the same response as above.)

Reviewer 3 Report

Dear Authors and Editors!

Thank you for the opportunity to review your manuscript.

It is intersting and useful topic - the pathogenesis of LN.

I have several suggestions

1) Please add more information about the role of cytokine in LN in the introduction

2) In the methods please add information about the distribution (normal or no), type of statistics and I think the Me (IOR) will be more convenient to present your data.

3) It is unclear when you collect the samples, because time SLE disease duration at baseline - 9.9 years. When patient had active LN, usually it occurs close to onset. You can provide the figure, explaining the timepoints.

4) F - females?

5) Complement consumption? Better use Low complement

6) Please use full names instead of abbreviations in the titles of the figures.

7) Please avoid CK or CKs. Use cytokines.

8) Add more limitations points in the subsequent part of the discussion

Author Response

(The authors gave the same response as above.)

Round 2

Reviewer 1 Report

The Authors included all the concerns. The manuscript is well supplemented and the preliminary nature of the research was clearly emphasized

Reviewer 2 Report

Thank you for the revision